# A predictive focus of gain modulation encodes target trajectories in insect vision

Steven D Wiederman[1]*[†], Joseph M Fabian[1][†], James R Dunbier[1], David C O'Carroll[2]

[1]Adelaide Medical School, The University of Adelaide, Adelaide, Australia; [2]Department of Biology, Lund University, Lund, Sweden

**Abstract** When a human catches a ball, they estimate future target location based on the current trajectory. How animals, small and large, encode such predictive processes at the single neuron level is unknown. Here we describe small target-selective neurons in predatory dragonflies that exhibit localized enhanced sensitivity for targets displaced to new locations just ahead of the prior path, with suppression elsewhere in the surround. This focused region of gain modulation is driven by predictive mechanisms, with the direction tuning shifting selectively to match the target's prior path. It involves a large local increase in contrast gain which spreads forward after a delay (e.g. an occlusion) and can even transfer between brain hemispheres, predicting trajectories moved towards the visual midline from the other eye. The tractable nature of dragonflies for physiological experiments makes this a useful model for studying the neuronal mechanisms underlying the brain's remarkable ability to anticipate moving stimuli.

*For correspondence: steven. wiederman@adelaide.edu.au

[†]These authors contributed equally to this work

**Competing interests:** The authors declare that no competing interests exist.

## Introduction

A diverse range of animals are capable of visually detecting moving objects within cluttered environments. This discrimination is a complex task, particularly in response to a small target generating very weak contrast as it moves against a highly textured background. The neural processing underlying this behavior must enhance a localized, weak and variable signal, which may only stimulate one or two photoreceptors in turn. Rather than simply respond reactively, some animals even anticipate a target's path by predicting its future location. In the vertebrate retina, high initial gain combined with neuronal adaptation and sensitization allows responses from a network of overlapping ganglion cells to 'keep up' with the current target location and account for sluggish neuronal delays (*Berry et al., 1999*; *Kastner and Baccus, 2013*). This encoding anticipates targets moving in a straight line, with trajectory reversals eliciting a synchronous burst of activity from a population of ganglion cells (*Schwartz et al., 2007*; *Chen et al., 2014*). However, this anticipation does not use the recent trajectory to extrapolate likely target locations at future times. Rather, the last observed location remains sensitized after the target disappears. This differs from studies of human observers, where a temporally occluded target results in improved sensitivity at the extrapolated forward location (*Watamaniuk and McKee, 1995*). This predictive encoding of future target locations indicates the presence of additional processing mechanisms beyond the retina.

Like human ball players, dragonflies also estimate target location, capturing single prey in visual clutter, even amidst a swarm of potential alternatives (*Corbet, 1999*). We recently described a 'winner-takes-all' neuron in the dragonfly likely to subserve such competitive selection of an individual target, whilst ignoring a distracter (*Wiederman and O'Carroll, 2013*). In other animal models, inhibitory circuits drive the selection of salient stimuli (*Mysore and Knudsen, 2013*) and the direction of attention towards targets is evidenced by modulation of contrast gain (*Moran and Desimone, 1985*; *Reynolds et al., 2000*). How prediction relates to the selection of salient stimuli is unknown

**eLife digest** Catching a ball requires a person to track the speed and direction of a small moving target often against a cluttered and varying background. Predatory insects, like dragonflies, face a similar challenge when they pursue their prey through the air. The task is made a little easier, however, by the fact that most moving targets tend to follow predictable trajectories. Indeed, animals are also better at tracking targets that follow smooth continuous trajectories, suggesting that brains have evolved to exploit the normal behavior of visual stimuli to reduce their workload

To find out how this process works, Wiederman, Fabian et al. studied the brains of dragonflies as they watched a black square intended to mimic prey. Brain cells called Small Target Motion Detectors (or STMD neurons for short) became more active in response to the target. But rather than simply following the target, the STMD neurons instead predicted its future location. In fact, individual neurons were more sensitive to movements occurring just ahead of the target's current position, and less sensitive to movements occurring elsewhere.

If the target abruptly disappeared, the point in space where the neurons were most sensitive to movement continued to gradually move forward over time. Given that real-life targets typically disappear when they move behind other objects, this suggests that the brain is predicting where the target is most likely to reappear. The STMD neurons became more sensitive to movement by increasing their ability to detect differences in brightness between the target and its background. In some cases, the neurons increased their sensitivity more than five-fold.

Insects and mammals last shared a common ancestor more than 500 million years ago, and, in many respects, mammalian brains are substantially more complex than insect brains. Nevertheless, the results of Wiederman, Fabian et al. show that the insect brain can perform visual tasks that were previously associated only with mammals. Neuroscientists and engineers have used the insect brain for decades to study the circuits that support biological processes. In the coming years, insects such as the dragonfly may enable us to explore how visual regions of the brain predict future events. This knowledge could ultimately be applied to artificial vision systems, such as those in self-driving cars.

(*Zirnsak et al., 2014*) and how selection, prediction and attention are encoded at the neuronal level is an intense topic of scientific investigation.

Here we used intact, in vivo, recordings from the system of small target-selective neurons in predatory dragonflies to reveal local changes in sensitivity elicited during target tracking. We show that this involves a large increase in contrast gain just ahead of the target's most recent location, with suppression in the surround. We investigated the spatial extent, temporal persistence and direction tuning within this region of enhancement. Our data shows that a local increase in gain spreads forward after a delay, even anticipating the path of primers presented to the contralateral eye and moved towards the visual midline. Moreover, the direction tuning shifts to match the prior path. Such response attributes differentiate this neuronal processing from typical models of direction selectivity and are ideally suited for a dragonfly's predictive pursuit of prey (*Mischiati et al., 2015*).

## Results

### Receptive fields are modulated by stimulus history

'Small target motion detector' (STMD) neurons in the dragonfly, *Hemicordulia tau*, are tuned to target size and velocity and are highly sensitive to contrast (*O'Carroll 1993*, *Wiederman et al., 2008*, *2013*). One identified STMD, CSTMD1, responds selectively to a small, moving target, even when embedded within natural scenes (*Wiederman and O'Carroll, 2011*). CSTMD1 also exhibits a sophisticated form of selective attention. The neuronal response to the presentation of two simultaneously moving targets does not simply result in either neuronal summation or inhibition. Instead, CSTMD1 responds in a winner-takes-all manner, selecting a single target as if the distracter does not even exist (*Wiederman and O'Carroll, 2013*).

We mapped CSTMD1's receptive field by measuring spiking activity in response to a single, black, square target (1.5°x1.5°) moving along trajectories at varying spatial locations in the visual

field. In one region, a gridded array (10 × 10) of short, vertical, target trajectories evoke weak neuronal responses (*Figure 1A*). For each short 200 ms trajectory, we plot mean spike rate over a 100 ms analysis window (from 50 to 150 ms). This colormap represents the spiking activity in response to short trajectories for each of the 100 corresponding spatial locations. In comparison, we present an identical square target moved along long, vertical trajectories (*Figure 1B*, *Video 1*) and segment

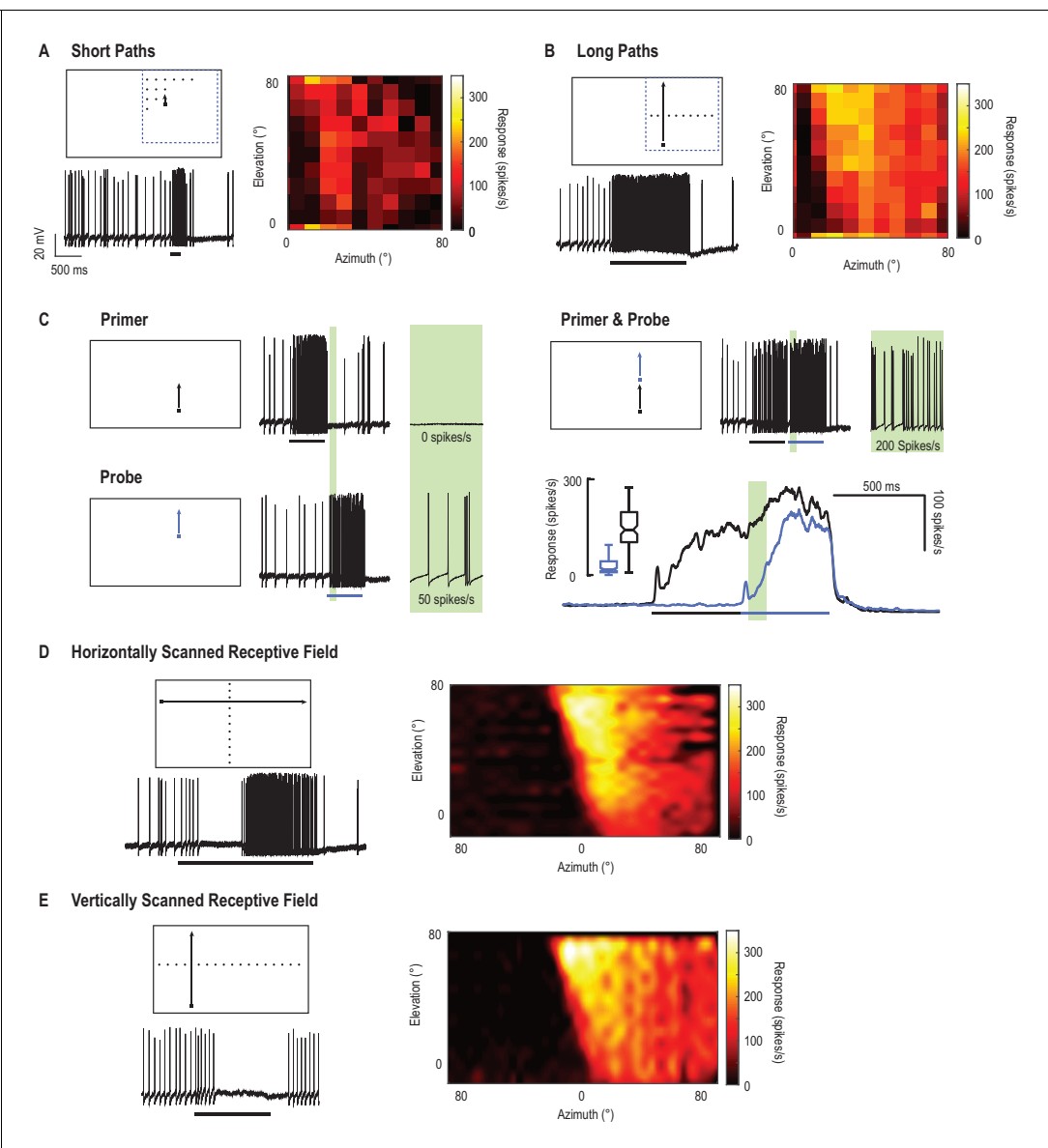

**Figure 1.** CSTMD1's receptive field mapped with drifting targets. (**A**) Small targets (black squares, 1.5°x1.5°) move along short trajectories (200 ms) that are both vertically and horizontally offset on a 10 × 10 grid. Pictograms are illustrative and not to scale. The colormap reveals CSTMD1 responses to these stimuli producing an 'unfacilitated' receptive field (50–150 ms analysis window). (**B**) Horizontally offset targets are drifted vertically up the monitor display along long, continuous trajectories eliciting strong, facilitated responses (100 ms bins to corresponding spatial locations in A). (**C**) Separating long paths into two components (primer followed by probe), allows us to examine the facilitatory effects within a short analysis window (before the probe self-primes, green region). In a single neuron, we examined response time courses (mean of 140 replicates over two hours) to repeated probe alone (blue line) and primer & probe (black line) conditions (**D**) We have previously described facilitated receptive fields in response to targets drifted across the entire visual display. Targets moving rightwards (vertically offset) reveal inhibition in one eye's visual field (in response to motion from the periphery towards the frontal area) and excitation in the other (from frontal to periphery). (**E**) The facilitated receptive field mapped with upwards moving targets (hot colors) is stronger than the weaker, though similarly shaped, unfacilitated receptive field in A. Targets moved upward in the other visual hemifield inhibit responses to below spontaneous levels (data trace).

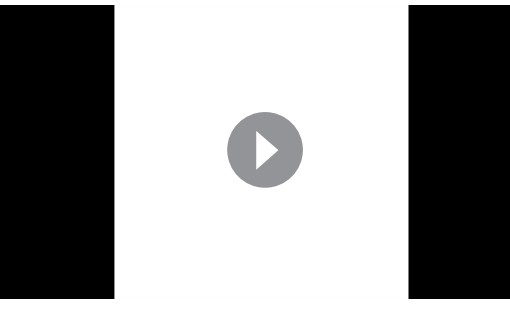

**Video 1.** Visual stimulus for *Figure 1*. The receptive field of CSTMD1 is mapped with a series of targets drifted on short paths (*Figure 1A*), or a single target drifting across the same location on a long path (*Figure 1B*). Separating a long target path into two components (a primer and a probe) allows us to quantify the facilitation induced by a primer (*Figure 1C*). All trials were presented in a randomised order. In this video trials are presented without rest periods, whilst in experiments trials were separated by at least 7 s to minimize habituation.

responses at the same corresponding spatial locations as the short paths in *Figure 1A* (mean spike rate over 100 ms bins). This reveals higher overall spiking activity in response to the long, continuous target trajectories. Here we investigate this effect of stimulus history by separating trajectories into components; a primer and a probe. Each elicit responses when presented alone, however, the probe's initial response is affected by the gain induced by a *preceding* primer (*Figure 1C*). We note that neuronal responses build slowly over hundreds of milliseconds - a property we have previously termed facilitation (*Nordström et al., 2011*). For the primer & probe condition (where a primer always precedes the probe stimulus) responses to the probe are facilitated (green region, cf. black with blue time courses). This facilitatory effect is not simply due to slow kinetics, as both responses have a rapid decay time course when the stimulus ends (*Dunbier et al., 2012*).

Previously, we have reported receptive fields in their facilitated state (*Dunbier et al., 2012*), mapped using targets moving along either long horizontal (*Figure 1D*) or vertical trajectories (*Figure 1E*). These reveal CSTMD1's excitatory receptive field which extends from the dorsal, visual midline to the periphery. Spatial inhomogeneity within this receptive field (interpolated to reduce binning artefacts) likely results from underlying dendritic integration and local spatiotemporal tuning differences. In the other visual hemifield (midline at 0° azimuth), a drifting target generates inhibition (*Figure 1D,E*), with activity suppressed to below spontaneous levels (0 spikes/s from a spontaneous activity of 11 ± 4 spikes/s, mean ± std.).

What is the effect on a 2D array of 'probe' responses (short paths in *Figure 1A*) when a long primer is presented along a single, *constrained*, trajectory immediately preceding each probe? Such an experiment would provide us with a snapshot of the effect of stimulus history (the primer target) on the current receptive field. *Figure 2* provides examples of individual, neuronal responses to short target trajectories (probes, blue arrows), both with and without a preceding 1 s duration target trajectory (primer, black arrow). For each probe location (in a 10 × 10 grid), we measured the spike rate within a 100 ms time window (the green shaded regions in *Figure 2*). The effect of priming was calculated as the difference (Δ spike rate) between the probe response when preceded by the primer ('primer & probe', black and blue arrows) and the probe alone (blue arrow) conditions. In this paradigm, we changed the spatial offset (jumps) between primer and probe without any delay (*Figure 2A,B*) or following a 300 ms pause (*Figure 2C,D*). We also tested a condition where the primer target drifted toward the dragonfly's midline, through the visual field of the other eye (*Figure 2E,F*). By ensuring primers did not enter the region of binocular overlap, any changes elicited in the probe locations (in the opposing eye) must have traversed brain hemispheres.

## Predictive modulation of gain

*Figure 3A* shows the complete two-dimensional map of primer-induced gain modulation, averaged across repeated intracellular recordings from CSTMD1 in different animals. Receptive fields are perspective-corrected from the dragonfly's point of view to a dragonfly eye map (mirrored along the vertical midline) and smoothed using bicubic interpolation to remove binning artefacts. The contour lines in *Figure 3A–C* indicate the average unfacilitated responses to the probe alone condition. In primer & probe trials, the primer target moved upwards (*Figure 3A,B*, *Video 2*) or rightwards (*Figure 3C*) along different paths in each trial but constrained within a region 5° wide (indicated by the white outlined box). To the CSTMD1 we recorded from (with its excitatory inputs located in the right mid-brain), upward and rightward moving targets represent progressive stimuli (i.e. moving from front-to-back). The small variation in primer path decreased local habituation from a repeating

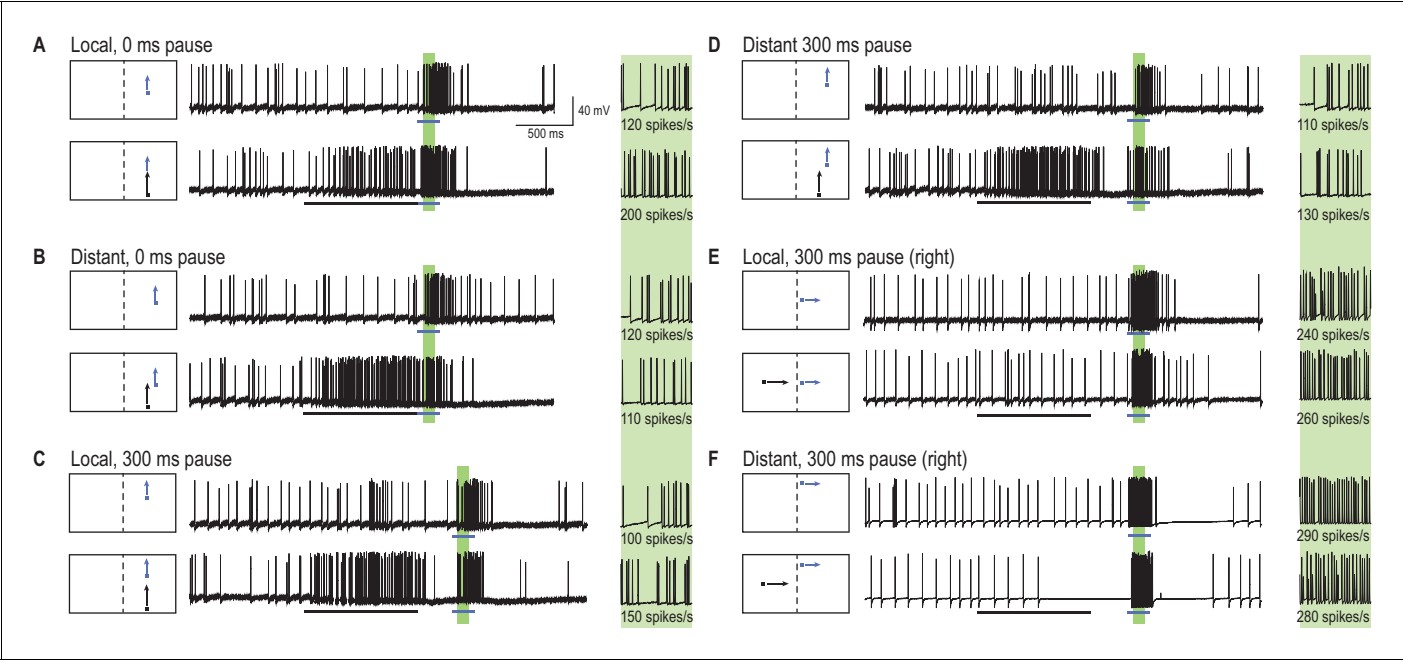

**Figure 2.** A primer target changes probe responses. (A) Example traces of CSTMD1's response to a probe target alone (blue arrow) or following a primer target (black arrow). The effect of the primer is measured as the difference (Δ spike rate) in response activity (primer & probe – probe alone) in the corresponding 100 ms window (green shaded region, with enlarged view on right). (B) With the primer spatially constrained, we repeat primer & probe and probe alone trials in a gridded array of 100 locations (200 trials in total, randomly interleaved). (C, D) A pause of 300 ms is inserted between the conditions where the primer disappears before probe onset (i.e. simulating a target occlusion). (E, F) A primer placed in the visual field of the other eye and moved toward the visual midline tests for information traversing the brain hemispheres.

primer running over the exact same trajectory. Probe alone and primer & probe trials were randomly interleaved. The color map reveals the average change in neuronal activity (Δ spike rate) elicited by the spatially constrained primer for each probe location (primer & probe – probe alone). *Figure 3A* reveals a pronounced 'focus' of increased sensitivity just ahead of the final location of the priming target and an extensive region of suppression in surrounding locations (mean, n = 9 dragonflies). Thus, what we have previously referred to as facilitation is a more complex phenomenon - local enhancement with spike rate *suppression* elicited by probes jumped into the surround. Here we use the term 'focus' to refer to both the local enhancement and widespread concomitant inhibition. Such neuronal processing may be indicative of an attentional mechanism, rather than a global arousal or sensitization (*Slagter et al., 2016*). In *Figure 3A*, we observed a large mean change in spike rate – over 50% increase within the focus center (p=0.0007, n = 9) and up to 50% decrease in surrounding locations (p=0.005, n = 9).

If the primer disappears for 300 ms before each probe, a similarly intense focus is still evident (*Figure 3B*, *Video 2*), but now spread forward in spatial extent (p=0.005, n = 7 dragonflies). The focus seems to account for the expected target location had it continued on its original trajectory (to a position as indicated by the white cross-hair in *Figure 3B,C*), albeit with an increased uncertainty given its broader spatial extent (mean, n = 7 dragonflies). Moreover, if we move a horizontal primer toward the visual midline in the contralateral eye before it disappears for 300 ms, the focus transfers across the brain to the ipsilateral hemisphere (*Figure 3C*). We then observed enhancement (red) localized to a broad region ahead of the primer trajectory (p=0.004, n = 7 dragonflies), but strong suppression (blue) at higher elevations (p=0.02, n = 7 dragonflies). Dragonflies have a small area of binocular overlap between the two eyes corresponding to the frontal/dorsal visual field (*Horridge, 1978*). Our stimulus was carefully designed to avoid this region, disappearing just before entering the area of overlap. Therefore, our result cannot be explained by facilitation being regenerated in the ipsilateral eye. Rather it must involve a localized, inter-hemispheric transfer of information. Furthermore, a localized and spatially segregated combination of enhancement and

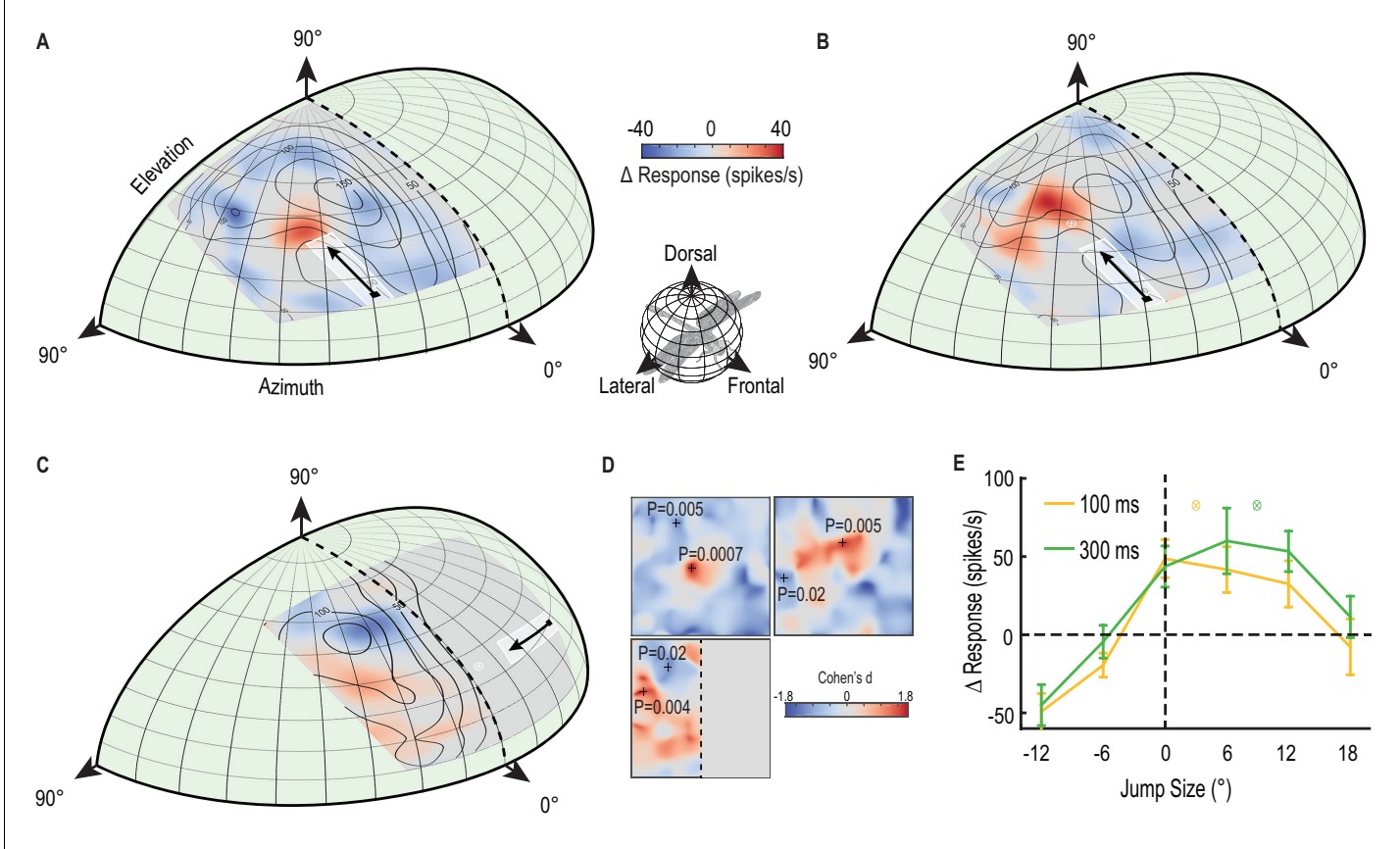

**Figure 3.** A predictive focus facilitates responses to a moving target. (**A**) The probe receptive field in response to short, vertical trajectories is indicated by contour lines (mean, n = 9 dragonflies). The color map shows change in spike rate (for each location) due to the immediately preceding primer trajectory that is presented within the white outlined box. The change in spiking activity in the corresponding analysis window reveals >50% enhancement in front of the moving target (red), but suppression in the surround (blue). (**B**) With a 300 ms delay introduced after the primer, the focus spreads forward (color map, n = 7 dragonflies), estimating the theoretical future target location (white crosshairs). (**C**) The primer moves toward the midline in the other eye's visual field, whilst avoiding binocular overlap. The focus transfers between brain hemispheres, with a spatially-localized enhancement in front of the target and suppression at higher elevations (color map, n = 7 dragonflies). (**D**) We examined the statistical significance of all three mappings (Figure A-C) by calculating the effect size at each spatial location (Cohen's d). We see values within the range ±1.8, well above those considered as large effect sizes (>0.5). For spatial points of interest (+), we calculate the corresponding statistical significance (P value) between the primer & probe and probe alone versions (**E**) There is a forward shift in the focus region (mean ± SEM, p=0.03, n = 12 dragonflies) following an occlusion (cf. 100 ms pause, yellow line with 300 ms pause, green line). The expected target locations following occlusions are indicated with color crosshairs (3° for 100 ms and 9° for 300 ms).

suppression (red and blue regions in *Figure 3C*) cannot be explained by a simple global mechanism, such as, a post-inhibitory rebound following a strong inhibitory stimulus (*Bolzon et al., 2009*). This transfer of a predictive focus between brain hemispheres is likely to play a crucial role in the prediction of target location during pursuit flights where the pursuer attempts to fixate the target frontally (*Mischiati et al., 2015*). Prolonged pursuit flights of conspecifics involve highly convoluted paths in which the target may readily cross from one visual field to the other (*Land and Collett, 1974*). In *Figure 3D*, we show the effect sizes of the three maps (*Figure 1A–C*) at all spatial locations. These Cohen's d values are the mean differences between primer & probe and probe alone (Δ spike rate), divided by the standard deviation of these differences. Cohen's d values over 0.5 are considered large effect sizes, thus our values of up to 1.8 in both excitatory and inhibitory directions are considerable. For particular points in these maps (*Figure 3D*, +'s) we calculate the paired, two-tailed P-values, highlighting the statistically significant effect of the primer.

*Figure 3E* shows data from an additional 12 dragonflies, where we mapped the forward-spreading focus following a delay of either 100 ms or 300 ms along a single dimension. These data (mean ±

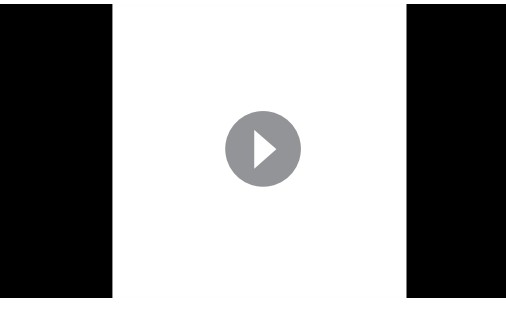

**Video 2.** Visual stimulus for *Figure 3*. The unfacilitated receptive field is mapped by a 10 × 10 grid of probes moving on short paths (*Figure 3A*, contour lines, only five trials shown in this video). A primer drifts on a long trajectory towards the center of the screen, before repeating presentation of each probe. Identical trials are replicated with a 300 ms pause separating the primer and probe (*Figure 3B*). All trials were presented in a randomised order, separated by rest periods of at least 7 s.

SEM) show that a small (6°) jump backwards precisely over the previously primed path already resets the response magnitude to that of the unfacilitated response (dashed line), whilst larger jumps backwards (12°) reveal potent suppression. Considering that the largest jump in this case is stimulating a part of the receptive field that last saw the target up to 700 ms earlier, the profound inhibition seen for this stimulus suggests that the prior primer target exerts long-lasting effects on the surrounding receptive field. Targets that jump forward after a delay reveal a shift in facilitation, spreading further forward after 300 ms (green line), compared to 100 ms (yellow line). Examining the mean difference combined across all forward jumps (6°, 12° and 18°) reveals a statistically significant difference between 100 ms and 300 ms pauses (p=0.03, Cohen's d = 0.7). Here the probe target followed directly 'on path' to the priming stimulus, without the small horizontal offsets (up to 5°, *Figure 3A–C* white priming region) used previously to limit local habituation.

In another eight dragonflies, instead of constraining the position of our primer, we instead tested responses to probes that always landed at the same location following different primers. This stereotyped probe followed either a jump in space, a pause in time, or a combination of both tests. *Figure 4A–D* show normalized response time-courses from individual CSTMD1 examples. The small 4° instantaneous jump ahead of the primer leads to a response time course with a very rapid rise to a level similar to the fully facilitated state (*Figure 4A*). However, a 12° instantaneous jump elicits a similar (slower) response time course to the unfacilitated probe (grey line), confirming the limited extent to which facilitation initially extends ahead of the target path. A large 20° jump ahead (*Figure 4A*) bypassing the focus-region entirely, again reveals surround suppression, with a much slower response time course than the control. Instantaneous backwards jumps (*Figure 4B*) also reveal potent suppression.

Pauses without a jump (0°), show that facilitation strength slowly decays over time at the last seen location of the target (*Figure 4C,E*, Cohen's d = 4.48). With no pause (0 ms), the strongest responses occur 4° in front of the moving target (*Figure 4E*) as observed in the 2D receptive fields (*Figure 3A*). Given that the target moves at 40°/s, it would have traversed 4°, 12° and 20° during pause 'occlusions' of 100 ms, 300 ms and 500 ms respectively. When larger jumps are matched to their respective pauses as might be expected during trajectory occlusions (12° and 300 ms; 20° and 500 ms) there is a statistically significant increase in the resultant spiking activity (*Figure 4E*, Cohen's d = 2.0 and 2.32 respectively).

## Primers increase contrast sensitivity
The data presented so far make a strong case for a complex predictive mechanism working to boost responses in a region where a target seen in the recent past is likely to move to in the near future. In primates, one known effect of such attentional or expectation effects is an upregulation of local contrast sensitivity (gain control) (*Reynolds et al., 2000*; *Carrasco et al., 2000*). To quantify changes in gain, we measured responses to varying contrast probes, preceded by either a low or high contrast primer (*Figure 5A,B*, *Video 3*). Both primers induced a large increase in response, with a larger output range (increased maximum response) and a greater than 5-fold increase in contrast sensitivity (*Figure 5C*, contrast threshold reduced from 0.071 to 0.013 for near threshold stimuli, $C_{50}$ from 0.36 to 0.13, n = 9 dragonflies). Lower contrast primers themselves induce less overall activity during the priming stimulus (*Figure 5D*, Cohen's d = 0.97), yet their effect on subsequent responses to stimuli presented at the expected location is remarkably similar to high contrast primers (cf. pink with red lines in *Figure 5C*). This suggests that the gain modulation is not elicited solely by the stimulus

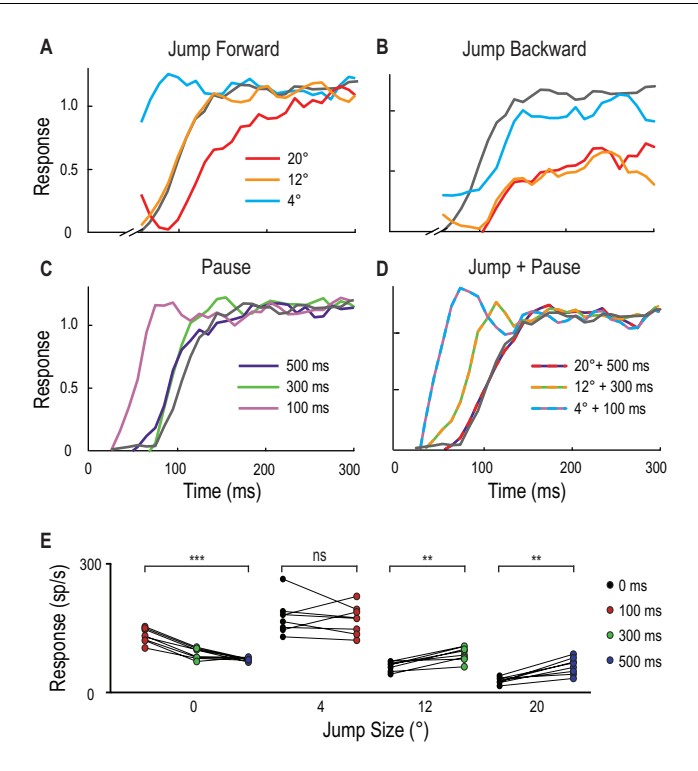

**Figure 4.** Spatial jumps and temporal pauses in target trajectories. (**A**) CSTMD1's normalized response to a short probe trajectory builds over several hundred milliseconds (grey line) and is changed by the position and timing of a 500 ms priming target. Probes jumped forward immediately following the primer, reveal kfacilitated responses (4° ahead), unfacilitated responses (12° ahead) and suppression (20° ahead), indicative of the focus-region in *Figure 3*. (**B**) A jump immediately back over the primer path exhibits unfacilitated (4° behind) or strongly inhibited (12° or 20° behind) responses. (**C**) Inserting a temporal pause between primer and probe shows that weaker facilitation persists at the primed location for over 500 ms, diminishing as the pause duration increases. (**D**) Combining a short pause with a jump reveals a forward spread of facilitation that could account for an occlusion. (A-D, n = 9 technical replicates from one dragonfly) (**E**) At the target's last seen position (jump size 0°), probe responses decrease at times following the primer's disappearance (p=0.0005). In comparison, responses to probes jumped 12° and 20° ahead increase when matched to their corresponding occlusion durations of 300 ms (p=0.008) and 500 ms (p=0.008). Asterisks indicate significance, n = 8 dragonflies.

contrast or the neuronal activity induced by the primer per se, but rather by target presence. This may indicate a 'switch' process, such as that suggested for neural circuits in the auditory brain stem of the barn owl (*Mysore et al., 2011*), rather than a simpler, activity-dependent gain control mechanism. Another interesting feature of the facilitated contrast sensitivity is that the boost of response gain is largest at mid-contrast, with softer saturation at high contrasts, extending the range of contrasts over which the response is modulated by a full order of magnitude. Both observations make sense considering the natural context for target detection. During pursuit flights, resources could thus be directed to the expected target location independent of its varying contrast as it moves across a cluttered background. Moreover, the reduction in slope of the contrast sensitivity function would reduce overall response variance to changes in the contrast of the selected target, a phenomenon also observed in humans (*Avidan et al., 2002*).

Our results also show that the increased contrast sensitivity is localized to the focus-region evident in *Figure 3*. A more distant primer displaced 20° to the side of the probe does not evoke facilitation of the contrast sensitivity function (*Figure 5C*). Instead, the contrast sensitivity function reveals a weaker effect of the surround suppression observed in the 2-D receptive fields (*Figure 3*). These contrast experiments used a shorter primer duration (600 ms vs. 1 s), suggesting that suppression could result in part from CSTMD1's global activity, rather than presynaptic processing. Facilitating

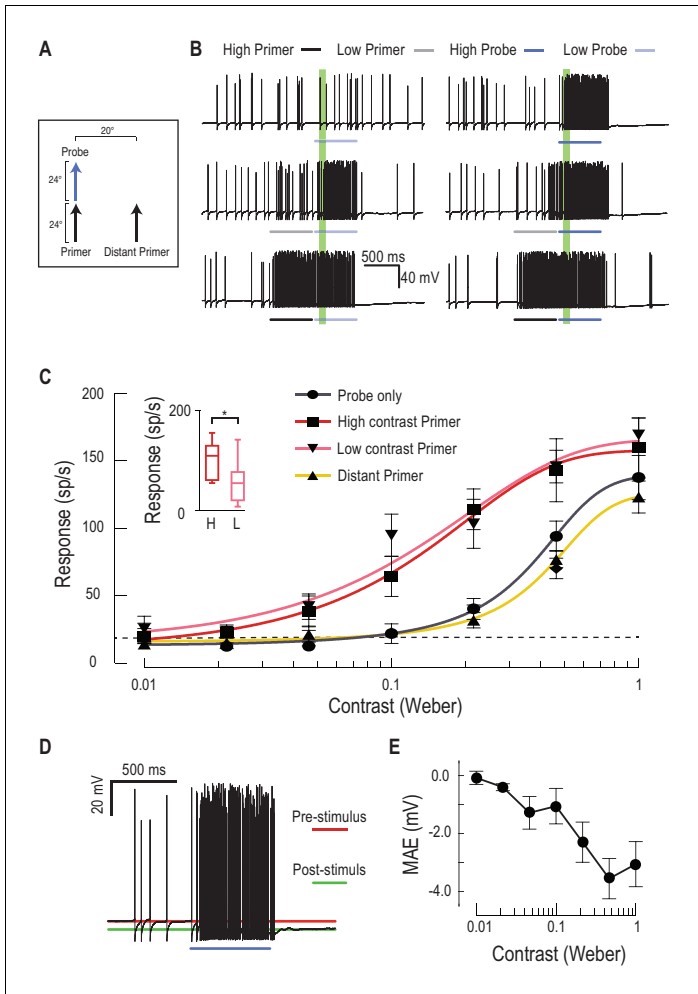

**Figure 5.** Low or high contrast primers increase probe contrast sensitivity. (**A**) Either a low or high contrast primer is presented before varying contrast probes (contrast sensitivity function). These either continue the path trajectory or jump to a distant location. (**B**) Example data traces of responses to either low (grey) or high (black) contrast primers that are presented before a series of varying contrast probes (light, medium and dark blue) (**C**) CSTMD1's sensitivity to varying contrast probes exhibits a sigmoidal function (grey), with the dashed line indicating a detection threshold above spontaneous levels. Following either a nearby low contrast (pink) or high contrast (red) primer, contrast sensitivity is substantially increased (n = 9 dragonflies, p<0.0001). A distant primer (yellow) does not elicit facilitation, even though spiking activity during low and high contrast primers (final 100 ms) is significantly different (inset, n = 9 dragonflies, p=0.02). (**D**) In response to an excitatory stimulus (e.g. high contrast stimulation), the underlying membrane potential is hyperpolarized, a form of motion-after-effect (MAE). (**E**) The hyperpolarizing motion-after-effect is related to the strength (e.g. target contrast) of the stimulus.

stimuli certainly increase the firing rate against a steadily hyperpolarizing membrane potential (*Figure 5D*). Following a high contrast primer, this hyperpolarizing motion-after-effect (MAE) reaches almost 4 mV and suppresses subsequent spiking activity for several hundred milliseconds (*Figure 5D,E*), an attenuation that may compete with spatially-localized facilitation. Interaction between the facilitation time course and longer-term suppression with slow kinetics may be analogous to the 'inhibition of return' observed in human reaction time experiments (*Posner and Cohen, 1984*).

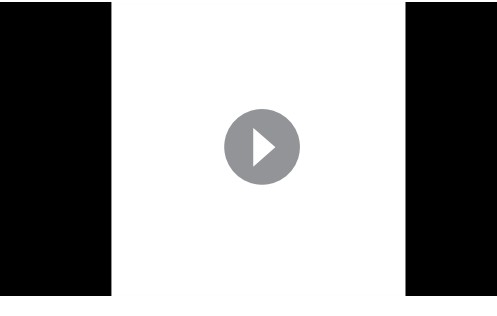

**Video 3.** Visual stimulus for *Figure 5*. Probes of varying contrast drift on short paths to determine unfacilitated contrast sensitivity (*Figure 5C*, only 1 contrast shown in video). Probes are then preceded by a high contrast primer that drifts at the same horizontal path as the probe - High Contrast Primed (local), or on a different horizontal path - High Contrast Primed (distant). Primed trials are also repeated with lower contrast primers (Low Contrast Primed). All trials were presented in a randomised order, separated by rest periods of at least 7 s.

## Primers induce directionality

The facilitated response of CSTMD1 appears to be only weakly direction-selective when stimulated with targets moving along prolonged paths. Within each hemifield, CSTMD1 has a weak preference for progressive motion upwards and away from the midline (rightwards for the neurons recorded here) (*Nordström et al., 2011*). To test whether the focus also anticipates the *direction* of a moving target, we presented a primer moving along one of four cardinal directions, followed by a probe that moves in eight possible directions (*Figure 6A,B*, *Video 4*). Probe responses alone are both weak and weakly direction selective (*Figure 6C*, grey dots). But all four primers facilitate responses maximally in the direction of the primer's path, shifting the direction tuning to match that of the primer (*Figure 6C*). The $b_1/b_0$ ratio is a measure of the strength of directionality which is similar for each of the conditions (*Figure 6D*). However, the magnitude of facilitation (*Figure 6E*) is considerably larger in CSTMD1's weak preferred, direction (upwards and rightwards for this hemisphere's CSTMD1). Such targets would be those moving away from the dragonfly's own heading (*Olberg, 1986*) with the mirror-symmetric CSTMD1 expected to exhibit directional preference to progressive targets moving upwards and to the left. This suggests that the preference of both the underlying tuning and the recruitment of facilitation may be linked to a control role in downstream processing of target trajectories for pursuit. Following a reversal of the target trajectory (*Figure 6F*, blue and purple lines), CSTMD1's response is strongly inhibited compared to the corresponding probe alone response (grey lines). This contrasts with findings in the vertebrate retina, where a subset of ganglion cells respond strongly and synchronously to motion reversals (*Schwartz et al., 2007*).

Does the recruitment of enhanced responses in the direction of travel represent an alteration of the direction selectivity in underlying local motion detectors, or does it result from the offset position of the focus of gain modulation located just ahead of the most recent target location (*Figure 3A*)? We tested this by jumping the probe stimulus 4° forward into the predicted center of the focus region (*Figure 7A*, *Video 5*). This stimulus induced much weaker direction selectivity (*Figure 7B*) than those that radiate away from the end of the same priming path ($b_1/b_0$ ratio of 0.32 vs. 0.50, p=0.04). Probes that reverse direction relative to the primer are not facilitated, except when the probe jumps 4° into the focus center (*Figure 7C*, Cohen's d = 3.90). Thus, the predictive focus of gain modulation is a spatial phenomenon, established by the past trajectory. This suggests that the apparent direction selectivity induced by primers is not due to any change in the local bias of underlying motion detectors to any one stimulus direction, but rather from the overall displacement of the focus *ahead* of the target location. Over a target's developing trajectory, direction selectivity (quantified here as vector magnitude) is established even more rapidly than the gain in the facilitated response (*Figure 7D*). The emergence of directional tuning raises the intriguing possibility that the modulation assists anticipation of target trajectories - promoting the expectation of a continued path. How such tuning matches closed-loop pursuits of the hawking dragonfly with its prey or conspecifics is not yet known.

## Facilitation in earlier retinotopic neurons

CSTMD1 is a higher-order neuron with inputs in the anterior optic tubercle, a midbrain output destination of optic lobe interneurons. We also tested for the facilitatory component of the predictive gain modulation in likely pre-cursor neurons: small-field (SF) STMDs located at an earlier stage of

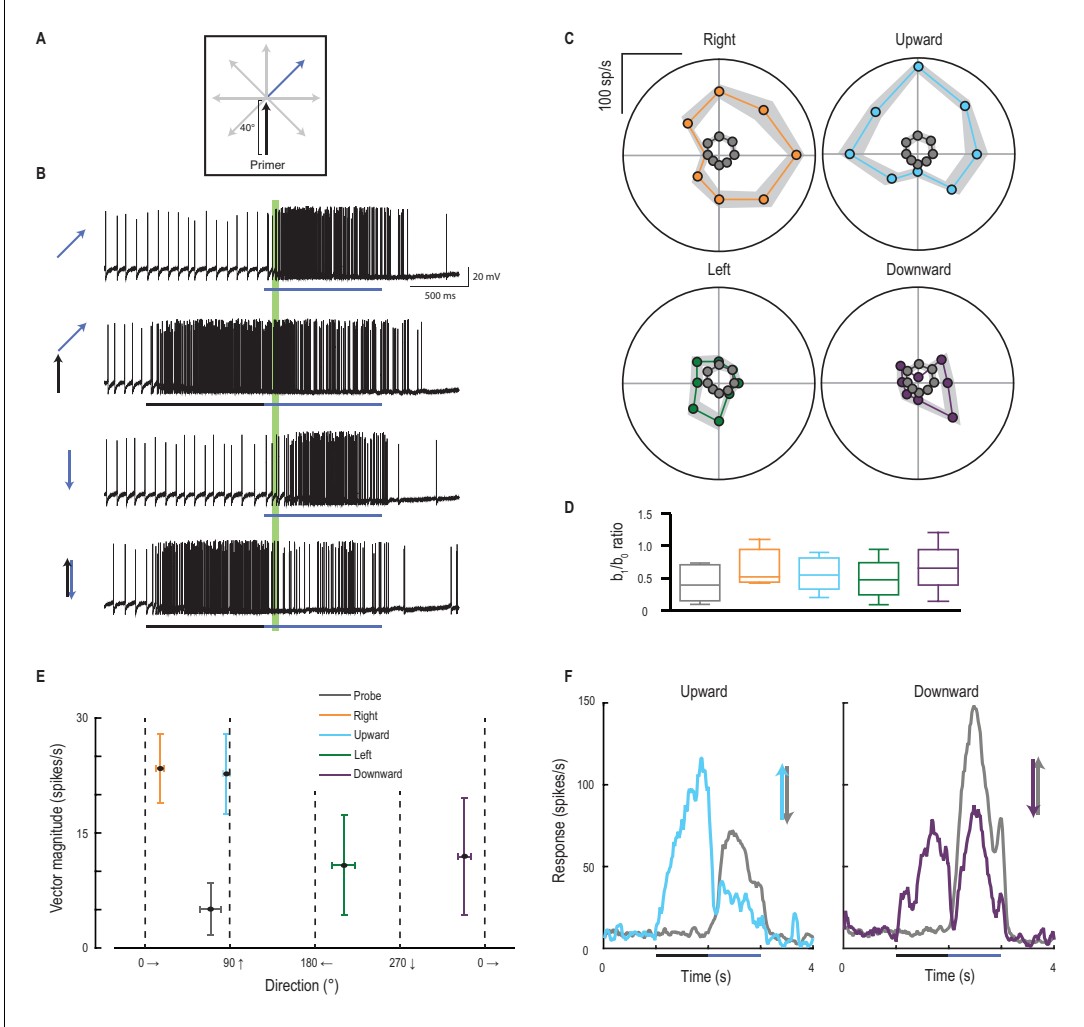

**Figure 6.** Primer direction establishes probe direction selectivity. (**A**) Primers of four possible directions (right, upward, left, downward) preceded probe responses in each of eight possible directions. (**B**) Examples of individual traces to a subset of the experiment conditions. The analysis period is indicated in green. (**C**) Probe responses are weak (grey points) until following a primer (in one of four cardinal directions) and are most facilitated in the primer's direction (mean ± SEM, n = 9 dragonflies). (**D**) The $b_1/b_0$ is an index showing the strength of directionality. (**E**) Polar plot vector magnitude and direction (mean ±95% CI), shows that probe direction selectivity generally aligns with the primer direction. (**F**) Either upward or downward probe alone (grey lines) evoke robust responses. However, 'reversals' (probes opposite in direction to a preceding primer) generate strong and long-lasting inhibition (mean time course, n = 9 dragonflies).

visual processing (*Barnett et al., 2007*). Retinotopically organized SF-STMDs have inputs in the outer lobula, a region akin to mammalian primary visual cortex (*Okamura and Strausfeld, 2007*; *O'Carroll 1993*). They have properties similar to end-stopped (hypercomplex) cells (*Nordström and O'Carroll, 2009*), which are modulated by contextual stimuli presented outside their classical receptive field (*Polat et al., 1998*). We presented primers outside SF-STMD receptive fields, that themselves induce no activity above spontaneous levels (*Figure 8A,B*, *Video 6*, n = 13 dragonflies), with probe stimuli that are limited to the classical (excitatory) receptive field. Primers moving toward the receptive field facilitate the probe responses by over 40%, whilst those heading away elicit no facilitation. This predictive gain modulation may be inherited and improved downstream, since we also observe facilitation in other large-field STMD neurons, with an average gain of over 80% (*Figure 8C*, Cohen's d = 1.02). Individual responses of both small and large field STMDs vary in facilitation strength, as well as overall activity. The retinotopic organization (*Figure 8D*) and facilitation

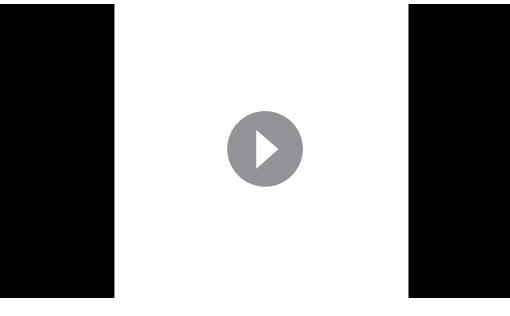

**Video 4.** Visual stimulus for *Figure 6*. Probes drift in 8 unique directions to determine the unfacilitated direction tuning of CSTMD1 (*Figure 6C*, only 4 directions shown in the video). The same 8 probes are preceded by primers moving in each of 4 cardinal directions (only upwards primer shown). All trials were presented in a randomised order, separated by rest periods of at least 7 s.

observed in SF-STMD's make them ideal candidates for mediating an interhemispheric transfer of localized predictive gain modulation. Supporting this hypothesis, at least one identified (dye-filled) SF-STMD axon traverses the brain with an output arborization located within a limited area of the contralateral lobula (*Figure 8D*). Neurons such as this are thus perfectly suited for the spatially localized inter-hemispheric modulation, both excitatory and inhibitory shown in *Figure 3C*.

## Discussion

Neuronal receptive fields are defined by their excitatory and inhibitory responses to stimulation. Populations of such responses elucidate network function, for example, as control systems in insect flight behaviour (*Gonzalez-Bellido et al., 2013*; *Maisak et al., 2013*). However, our results show that in addition to stimulus selectivity (contrast, size, velocity), a neuron's receptive field is also a dynamic representation of the spatial (*Wiederman and O'Carroll, 2013*) and temporal context. Here modulation of the dynamic receptive field represents anticipatory coding, a more complex influence of history than simple neuronal adaptation, sensitization, habituation or fatigue. Indeed, such complexity in processing is also evident in the 'omitted stimulus response' in the vertebrate

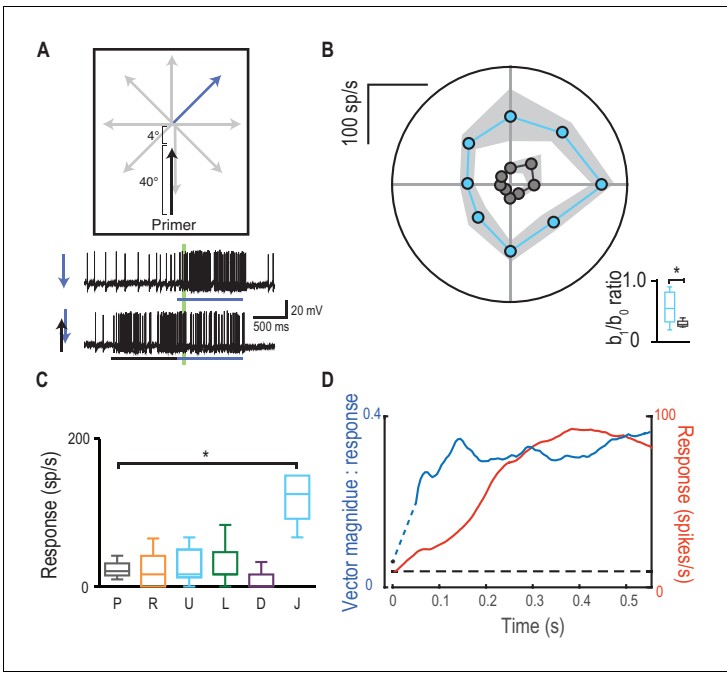

**Figure 7.** Direction selectivity is a result of spatial facilitation. (**A**) The direction experiment is repeated, now with a 4° jump forward into the spotlight. (**B**) Responses are facilitated for all directions (mean ± SEM, n = 5 dragonflies) with decreased direction selectivity ($b_1/b_0$). (**C**) Probes in the opposite direction to their corresponding primer reveal no facilitation or inhibition, except when jumped [J] into the spotlight (p=0.03). (**D**) The magnitude of direction selectivity builds on a faster timescale than the response onset.

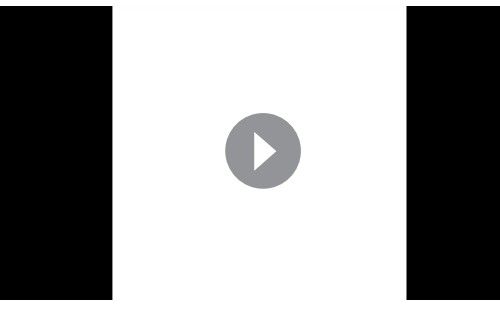

**Video 5.** Visual stimulus for *Figure 7*. Probes are presented in an identical manner to *Figure 6C*. However, here probes are preceded by a vertical primer that terminates 4° below the probe start location (*Figure 7B*). All trials were presented in a randomised order, separated by rest periods of at least 7 s.

retina, where an omitted component of a periodic pattern predictively elicits robust neuronal activity (*Schwartz et al., 2007*). These examples highlight that the brain is a 'predictive machine' (*Rao & Ballard 1999*). However, instead of encoding novelty or the unexpected, STMD neurons predict consistency of a selected target's trajectory, all whilst suppressing distracters.

Direction selectivity is, in effect, a simple form of prediction. For example, the Hassenstein-Reichardt correlator provides a nonlinear, facilitated response when an adjacent point is stimulated within a future period (*Hassenstein and Reichardt, 1956*). However, such direction selective models cannot explain the observation of a traveling gain modulation that spreads further forward, the longer the occlusion. Neither can these models account for changes in preferred direction, determined by the target's previous direction of travel. Such models do not result in a contrast invariant 'switch' establishing the focus strength, nor the presence of a large suppressive surround. Furthermore, the effects described here are on larger scales either spatially (tens of degrees) or temporally (hundreds of milliseconds) compared with local motion detection processes, such as optic flow analysis (tens of milliseconds, *Guo and Reichardt, 1987*). Finally, our results show a local, predictive focus of facilitation that traverses across brain hemispheres, which is an attribute more reminiscent of higher order attentional networks, rather than local motion encoding circuitry.

Our findings of over a 400% increase in contrast sensitivity is consistent with studies that cue spatial attention in vertebrates, albeit with a significantly larger increase. For example, the contrast gain of human observers is increased by approximately 40% for stimuli presented at an attended location (*Carrasco et al., 2000*), with concurrent decreased contrast gain for stimuli presented elsewhere (*Pestilli and Carrasco, 2005*). Similar results are also observed in single unit recordings from macaque V4, where gratings presented at attended locations elicit responses equivalent to a 51% increase in stimulus contrast (*Reynolds et al., 2000*). Whilst there is ongoing debate over whether attention modulates contrast gain (*Reynolds et al., 2000*) or response gain (*Lee and Maunsell, 2010*), the facilitation in CSTMD1 reveals a combination of both (*Figure 5C*). CSTMD1's gain modulation could be an inherent component of the prediction mechanism, or the result of the priming target acting as a cue directing attention to the targets predicted location.

We have previously reported that CSTMD1 selectively attends to one target when presented with a pair of competing stimuli, completely ignoring the distracter (*Wiederman and O'Carroll, 2013*). In repeated trials, the selected target was not always the same and even occasionally switched midtrial. This raises the intriguing possibility that the predictive focus 'locks on' to a single target, suppressing distracters. The anticipatory gain control measured here provides a possible explanation for this behavior – a positive feedback that allows the neuron to lock onto a single object while other mechanisms, including global inhibition, may help suppress competing objects. Future experiments will address the parameters of the stimuli (e.g. timing, salience) that permit the predictive focus to switch between alternative targets. Furthermore, we are currently investigating whether the predictive focus and competitive selection is elicited bottom-up by the stimuli (exogenous) or includes a top-down component (endogenous). That is, for a dragonfly feeding in a swarm, are target saliency attributes driving pursuit selection, or is the dragonfly choosing its prey from more complex internal workings?

For decades, scientists studied the neuronal basis of 'elementary motion detection' in true flies (*Diptera*). With morphological (*Takemura et al., 2013*) and physiological (*Maisak et al., 2013*) experiments making significant progress at elucidating this circuitry, increasing attention is now shifting towards other visual tasks such as feature discrimination (*Aptekar et al., 2015*; *Keleş and Frye, 2017*). Until now, there has been a divide between such 'simple' visual operations and higher-order

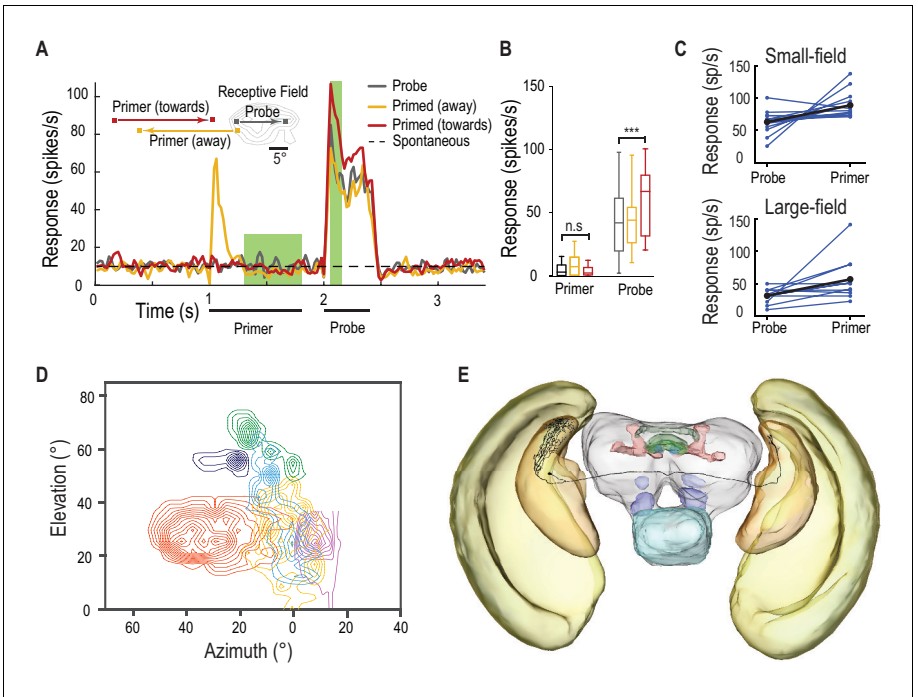

**Figure 8.** SF-STMDs are facilitated by a primer that moves toward the receptive field. (**A**) Primers move either toward (red) or away (yellow) from the classical receptive field (RF), preceding a probe target within the RF (mean, n = 13 dragonflies). (**B**) Outside the receptive field, primer responses do not significantly differ from spontaneous activity. Primers that move towards the receptive field increase probe responses by over 40% (p=0.0004, n = 13 dragonflies). (**C**) Individual STMDs, with either small or large receptive fields, exhibit varying degrees of facilitation (blue). Mean facilitation (black) increase responses by over 40% in small-field (n = 13 dragonflies), 80% in large-field STMDs (n = 11 dragonflies) and 50% in CSTMD1 (data not shown). (**D**) Six small-field STMD receptive fields (RF) are predominantly fronto-dorsal and exhibit variation in overall size and spatial locations. Contour lines represent 25 spikes/s. The SF-STMD with light purple contours is the same neuron in E, with inputs in the binocular region of the dragonfly's right visual field, whilst input dendrites are in the left hemisphere (**E**) An SF-STMD's axon traverses the brain, potentially underlying transfer of local predictive gain modulation.

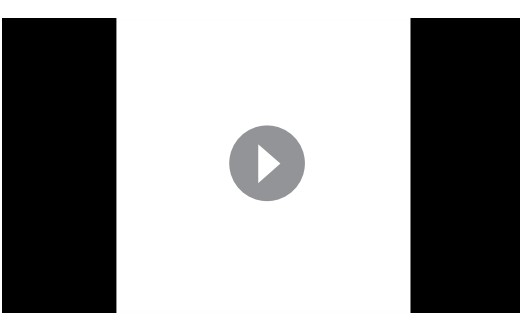

**Video 6.** Visual stimulus for *Figure 8*. Probes are presented within the receptive field of a Small-Field STMD neuron. Probes are preceded by primers that either drift towards the receptive field, or away from the receptive field (*Figure 8A*). All trials were presented in a randomised order, separated by rest periods of at least 7 s.

processing observed in mammals. Our results reveal the dragonfly as a surprisingly sophisticated, yet tractable model, permitting investigation of fundamental physiological and morphological principles underlying neuronal prediction and selective attention.

## Materials and methods

### Electrophysiology

We recorded from a total of 63, wild caught male dragonflies, *Hemicordulia*. Animals were immobilized with a wax-rosin mixture (1:1) and fixed to an articulating magnetic stand. The head was tilted forward and a small hole dissected in the posterior surface, exposing the left optic lobe.

We pulled Aluminium silicate electrodes on a Sutter Instruments P-97 electrode puller, and backfilled them with 2M KCl solution. Electrodes

were inserted through the neural sheath into the proximal lobula complex using a piezo-electric stepper (Marzhauser-Wetzlar PM-10), with typical resistance between 50–150 MΩ. Intracellular responses were digitized at 5 kHz with a 16-bit A/D converter (National Instruments) for off-line analysis with MATLAB.

Freshly penetrated cells were presented with small targets, bars and wide-field gratings for classification. Neurons were classed as STMDs when responding robustly to visual stimuli composed of small, moving targets and not responsive to bars or gratings. CSTMD1 was identified by its characteristic receptive field, response tuning and action potentials. STMD neurons were categorized into small or large-field by mapping their receptive fields with drifting targets (a half-width either less than, or greater than 25°).

### Visual stimuli and data analysis

We presented stimuli on high definition LCD monitors (120 Hz and above). The animal was placed 20 cm away and centered on the visual midline. Contrast stimuli were presented at screen center to minimize off-axis artefacts. Stimulus scripts (https://github.com/swiederm/predictive-gain) were written using MATLAB's Psychtoolbox and integrated into the data acquisition system (*Wiederman et al., 2017*; a copy is archived at https://github.com/elifesciences-publications/predictive-gain). Unless stated otherwise all targets were 1.5°x1.5° black squares drifted at 40°/s. A minimum of 7 s rest between trials was implemented to avoid habituation or facilitation from prior trials. Data were only ever excluded due to pathological damage of the neuron or extensive habituation (experiment cessation). All means are calculated from biological replicates (i.e. repeated measurements from identified neurons in different animals). Each biological replicate represents the mean of between 1 and 10 technical replicates.

### Statistical tests

We report exact P (unless miniscule). Due to the small sample sizes, all tests are nonparametric, two-sided and account for multiple comparisons. All box and whisker plots indicate median, interquartile and full minimum-maximum range (whiskers).

Predictive focus of gain modulation: We mapped the spatial extent of this focus with a series of 200 ms probe targets randomly presented across a $10 \times 10$ grid of locations within CSTMD1's excitatory receptive field. We calculated spike rate within an analysis window (50–150 ms) following probe onset at each location. We randomly interleaved unprimed probe stimuli with (in 50% of trials) corresponding probes that followed a 1 second-long primer target (n = 9 dragonflies). Priming targets moved vertically up the screen at 32°/s, pseudo-randomly presented within a 5° wide region (white outlined box) to minimize local habituation induced by the primer. For each spatial location (100 in total), we calculated the difference between probe response (following primer) with probe response (no primer). Inter-trial and inter-neuronal noise in the focus colormaps was reduced by averaging across dragonflies interpolated and slightly smoothed (Gaussian, σ = 0.5) matrices. This noise-reducing method effectively portrays the result of adding the primer target, however, may slightly blur the focus region due to averaging across samples. A second experiment followed the same protocol except with a 300 ms pause inserted between primer and probe (n = 7 dragonflies). A third experiment had the primer moving horizontally within the visual field of the contralateral eye with care taken to avoid the frontal region of 10° binocular overlap. This experiment also included a 300 ms pause inserted between primer and probe (n = 7 dragonflies). To examine the significance of these maps, we created Cohen's d versions. For each spatial location (*loc*), we calculated the mean difference (primer & probe – probe alone) across the sample size (n, number of dragonflies) and divided by the standard deviation of these differences across the sample size. This effect size represents the mean observed change at each location (across dragonflies) normalized by the standard deviation at each location (across dragonflies). Note that to avoid divide by zero errors, we did not calculate the Cohen's d values for the inhibitory hemifield (map for *Figure 3C*) which has no activity, and thus minimal standard deviation. For specific locations of interest, we tested for statistical significance by calculating the paired t-statistic and two-tailed P-value.

$$\mu_{loc} = \sum_{i=1}^{n} \left( (\text{primer \& probe})_i - (\text{probe alone})_i \right) - \mu)^2$$

$$\sigma_{loc} = \sqrt{\frac{1}{n-1}\sum_{i=1}^{n}\big((\text{primer \& probe})_i - (\text{probealone})_i\big) - \mu\big)^2}$$

$$Cohen's\ d_{loc} = \frac{\text{u}_{loc}}{\sigma_{loc}}$$

$$t_{loc} = Cohen's\ d_{loc} \cdot \sqrt{n}$$

Further experiments were conducted along a 1-dimensional path. Firstly, with a constrained location of the primer (n = 12 dragonflies) and then with a constrained location of the probe (n = 8 dragonflies). For the time courses, we normalized responses by dividing by fully-facilitated controls (corresponding spatial locations), thus accounting for spatial inhomogeneity in the receptive field. Statistical comparisons applied either Wilcoxon or Friedman's tests with multiple comparisons.

## Contrast sensitivity

We varied the contrast (7 values) of a probe target drifted upwards through CSTMD1's receptive field (n = 9). Target contrast (Weber) was defined as:

$$c_w = \frac{I_{target} - I_{background}}{I_{background}}$$

Probes drifted upwards for 600 ms at two possible locations separated horizontally by 20°. For each probe contrast, we measured responses in an analysis window (50–150 ms) following onset. Primers drifted upwards for 600 ms, either towards the probe location (primer) or displaced 20° to the side (distant primer). Primer contrast was either 1.0 (high contrast) or 0.2 (low contrast). Primer responses were quantified over the last 100 ms of primer motion. We inserted a 50 ms pause between primer and probe and primer to ensure that the residual primer response was not attributed to the probe. Trial order across all contrast sensitivity experiments was randomized. The parameters (top, bottom, logIC50 and hill-slope) of each contrast sensitivity function were compared with an extra sum-of-squares F test, whilst responses to primers were compared by Wilcoxon test. To define a detectability threshold for estimating contrast sensitivity, we measured spontaneous activity in the 1 s pre-stimulus period across all 630 trials (n = 9 dragonflies). For each neuron, responses were binned (20 ms) before calculating the upper 95th percentile of binned responses.

## Direction selectivity

We measured CSTMD1's direction selectivity by drifting probes in 8 directions from a central point in CSTMD1's excitatory receptive field (n = 9). Primers drifted for 1 s in each of 4 cardinal directions terminating at the probe location. In a separate experiment, vertically drifted primers terminated 4° below the probe location, placing all probes within the center of focus (n = 5). Probe responses were analyzed in a window (40–100 ms) following probe onset. This window is shorter and earlier than in other experiments to account for the rapid establishment of probe direction selectivity.

We quantify direction selectivity in two ways. We regress responses onto the sinusoidal model $R(\theta) = b_0 + b_1 \sin(\theta + \varphi)$, where R is the response at direction $\theta$, $b_0$ is the offset, $b_1$ is the directional component of the response and $\varphi$ is the phase (preferred direction). We also quantify the mean polar vector for each condition, calculating 95% confidence intervals for both vector direction and vector magnitude across all cells.

Differences in $b_1/b_0$ ratio between trials that were primed upward, and upward following a forward jump were compared with a Mann-Whitney test. The variance of responses to targets that turn back in the opposite direction to the primer were analyzed by a Kruskal-Wallis test, followed by Dunn's multiple comparisons. We applied the Kruskal-Wallis test because the direction tuning data with (*Figure 7C*, condition J) and without (*Figure 7C*, conditions P U R L D) a forward jump was obtained from different independent samples.

## SF-STMD facilitation

Responses were elicited by 400 ms probe trajectories, commencing motion within the classical receptive field and drifted in the neuron's preferred direction. Primers were either drifted for 800 ms 'towards' the excitatory receptive field, or from within the excitatory receptive field moving 'away'. Primers terminated at least 8° away from the classical receptive field, and were followed by a 200 ms pause before the appearance of the probe stimulus. Primer responses were analyzed in a window 300–800 ms following onset, whilst the probe was 50–150 ms following probe onset.

The remaining large-field and small-field experiments were performed across populations of neurons with varying overall activity. To normalize, neuronal responses for a given neuron were divided by a factor equal to the neuron's mean response to probes across all priming conditions. To convert responses back into spikes/s, we multiplied the normalized response by the mean factor for all neurons in the dataset. As all conditions were paired across independent samples, we compare responses across conditions by a Friedman test, followed by Dunn's multiple comparisons. All statistical tests presented are two-tailed.

## Dye filling

The morphology of an SF-STMD neuron was visualized by intracellular labelling with Lucifer Yellow. Iontophoresis was achieved by passing 3nA negative current through electrodes tip-filled with 4% Lucifer Yellow solution in 0.1M LiCl. Brains were then carefully dissected, fixed overnight in 4% paraformaldehyde at 4°C, dehydrated in ethanol series (70%, 90%, 100%, 100%), cleared in methyl salicylate and mounted on a cavity slide for fluorescence imaging.

## Data availability

Data obtained is managed per the ARC/NHMRC Australian Code for the Responsible Conduct of Research. Raw data from experimental testing and numerical simulation is stored on a locally managed server. Processed experimental and numerical data is available on the data management server for The University of Adelaide (https://adelaide.figshare.com)

## Acknowledgements

This research was supported by the Australian Research Council (DP130104572, DE150100548) and the Swedish Research Council (VR 2014–4904). We thank the manager of the Adelaide Botanic Gardens for allowing insect collection and behavioral recordings.

## Additional information

### Funding

| Funder | Grant reference number | Author |
| --- | --- | --- |
| Australian Research Council | DE150100548 | Steven D Wiederman |
| Vetenskapsrådet | VR 2014-4904 | David O'Carroll |
| Australian Research Council | DP130104572 | David O'Carroll |
| Swedish Foundation for International Cooperation in Research and Higher Education | STINT 2012-2033 | David O'Carroll |

The funders had no role in study design, data collection and interpretation, or the decision to submit the work for publication.

### Author contributions

SDW, DCO'C, Conceptualization, Resources, Data curation, Software, Formal analysis, Supervision, Funding acquisition, Validation, Investigation, Visualization, Methodology, Writing—original draft, Project administration, Writing—review and editing; JMF, Conceptualization, Resources, Data curation, Software, Formal analysis, Validation, Investigation, Visualization, Methodology, Writing—

original draft, Writing—review and editing; JRD, Conceptualization, Resources, Data curation, Software, Formal analysis, Validation, Investigation, Visualization, Methodology, Writing—review and editing

## Author ORCIDs

Steven D Wiederman, http://orcid.org/0000-0002-0902-803X
Joseph M Fabian, http://orcid.org/0000-0001-6739-7581
David C O'Carroll, http://orcid.org/0000-0002-2352-4320

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
