## [Decision Letter]

Thank you for submitting your article "An attentional spotlight predictively facilitates target trajectories in insect vision" for consideration by *eLife*. Your article has been favorably evaluated by Timothy Behrens (Senior Editor) and three reviewers, one of whom, Fred Rieke, is a member of our Board of Reviewing Editors.

The reviewers have discussed the reviews with one another and the Reviewing Editor has drafted this decision to help you prepare a revised submission.

All three reviewers agreed that the core results in the paper were interesting and that the experiments appeared to be well conceived and executed. The reviewers also agreed that several issues needed work before a complete evaluation of the paper could be made. Three issues in particular emerged from discussions among the reviewers. These are highlighted below, and are elaborated in comments from the individual reviewers.

1) Attention vs. prediction

A key point emphasized in the reviews is that the phenomena described do not clearly satisfy the requirements for attentional effects. Specifically, attention is defined relative to behavior and not neural responses. The effect described in the paper would seem better described as a "predictive gain enhancement" or "anticipatory coding" or something similar. See comments from reviewers 2 and 3 for more details.

2) Novelty and relation to direction selectivity literature

There is a large literature on direction selectivity – including some in retina which considers complex stimulus trajectories (particularly work from Michael Berry – e.g. Schwartz et al., 2007; Chen et al., 2013; Chen et al., 2014). This work reveals a number of non-classical features of directional coding. This literature seems the proper context in which to evaluate the novelty of the present work, and the paper would benefit from comparing the work with what is known about direction selectivity in other systems.

3) Presentation

Several issues made the paper harder to read than necessary. Most notably, the figure panels used to illustrate the stimuli were less helpful than they could be. It is essential to make this as clear as possible to the reader. See reviewer comments for specifics.

*Reviewer #1:*

In this paper, Wiederman and colleagues describe an interesting property of responses of visual neurons in dragonfly in which the sensitivity is enhanced to predict the trajectory of moving objects. I am not an expert in the main areas covered by the paper, but found the phenomena described quite interesting. There were a few places I felt the results could be presented more clearly:

The Results would likely to be easier to read with titled subsections that provide an outline of how the results are organized.

Figure 1) Indicate which stimuli correspond to the example traces shown. 2) Why are some receptive fields smoothed and others not?

Figure 2: Can you illustrate the differences in responses with and without the primer more clearly (e.g. blow up responses at the time of interest, perhaps give spike count as well)? In the present format it is not possible to see the differences in response, which means that Figure 2 supports Figure 3 less than it could.

Figure 5: Can you make the stimulus in B clearer or include a bit more in the figure legend?

The Discussion is quite brief. If possible, the Discussion could be strengthened by making more connections to the broader attention literature. For example, the specificity of the enhanced sensitivity shown here, and the shift in direction tuning, are quite interesting. Is there precedent in other systems for those?

Please include significance tests, e.g. for the data in Figure 3.

*Reviewer #2:*

Wiederman and colleagues present a nice set of data showing that the lobula neuron CSTMD1 in dragonflies changes its gain in anticipation of the location of a small moving object. This change in gain is targeted in space, and does not require temporally continuous stimuli. The authors carefully measure the changes in gain as functions of positional and temporal offsets, and also show that the similar anticipatory gain changes appear in small- and large-field STMD neurons, which are earlier in the dragonfly's visual system.

The experiments are expertly done and the effect appears well-characterized by the set of experiments presented here. This study demonstrates a very nice example of anticipatory gain enhancement by a small visual circuit, which relates well to other properties of this neuron and its potential role in dragonfly prey capture.

1) In the last sentence of the first paragraph, the authors suggest that retinal processing "does not predict an object's future location." However, in Kastner and Baccus, cited one sentence before, those authors state, "Sensitization acts to predict the future location of an object". The authors should clarify and expand their claims about what the retinal literature says and how it relates to the effects measured here.

2) The largest question I have about this work is whether the observed effects constitute an "attentional spotlight", as the authors contend, or whether it is a nice example of non-linear stimulus-stimulus interactions over space and time, which would not typically be thought of as attention.

A) Attention can be a squirrelly concept, but I think a prime place to look is in the retina literature, which seems not to have adopted the term "attention" for similar phenomenology, in which anticipatory coding is observed (Kastner and Baccus, 2013; Schwartz… Berry, 2007). Are the retina folks just being too timid in their interpretation of their data? The term favored here, 'spotlight of attention', appears to most often refer to top-down attention. For typical experiments showing top-down attention, experimenters present a cue to guide where gain modulation is focused, but crucially, that cue should be rather separate from the probe stimulus. (For instance, with moving colored dots, a cue to indicate whether color or direction is the important variable for a given trial's reward.) Without a behavior in these experiments, it seems very difficult to argue for top-down attention. The authors rightly view the gain changes they measure as stimulus driven – they refer to a primer stimulus throughout.

B) Bottom-up attention is much harder to distinguish from stimulus-stimulus interactions, and seems less amenable to the term "attentional spotlight". That term has been used to apply to serial searches in human psychophysics where stimuli do not 'pop out' (in Anne Treisman's work in the 1980s). In those experiments, the attentional spotlight is in contrast to the popping out of stimuli, and is inferred from the serial observation of each object in a set.

C) Note also that the human reference (Watamaniuk and McKee) for this type cognitive ability (tracking continuous motion in the face of occlusion) does not refer to attention. In the primate and human studies of attention cited in the discussion (Reynolds et al., 2000 and Carrasco et al., 2000), the visual cues that were used to direct the covert attention of the subject were quite distinct from probe stimulus; that seems very different from the priming cues in this study, which were short dot motions closely related to the longer probing stimulus.

D) The authors' previous work (Current Biology, 2013) refers to stimulus selection by this neuron as "selective attention", but that seems less loaded than "attentional spotlight". A spotlight of attention could exist in invertebrates; I'm just not sure this study shows evidence for it.

3) In Figure 3 and results presentation (Results, fifth paragraph), the authors suggest that the spread forward in gain is further when the delay is longer. By eye, this does not look significant, but the authors should test for significance to make this claim. It would also be nice to note the extrapolated position from the priming motion after the 100 ms and 300 ms pauses.

4) These authors previously reported that this neuron's activity switched back and forth between the activity related to the trajectory of just one of two simultaneously presented targets (Wiederman & O'Carroll, 2013). The anticipatory gain control measured here gives a beautiful explanation for positive feedback that could cause the neuron to lock onto a single object while suppressing competing objects. I think this would be a relevant point to add the Discussion, and could also be used to tie these effects back to dragonfly ethology.

*Reviewer #3:*

Here, Wiederman et al. study the responses of a particular motion-selective neuron (CSTMD1) in the dragonfly in the context of a moving probe stimulus. It was already known that these neurons are motion-selective. The novel part of this study is that it tests this motion selectivity in a number of new contexts, and shows that the "predictive" abilities of this neuron are quite impressive. These neurons have a gain that follows dynamically ahead of the path of a stimulus, with a number of key features: It (1) crosses the hemifield boundary, tracks the spatial and temporal expectation based on the stimulus velocity, and continues even if the target disappears briefly. The authors explicitly test the effect of this gain on stimuli of varying contrast, and find that it matches qualitatively well with contrast gain changes with attention seen in mammalian neurons in attention tasks.

Overall, this work is quite interesting and shows that these particular neurons in the dragonfly brain are doing some remarkable computation. The authors frame this work in the context of attention. This I find to be a dubious choice, and I think detracts from the work rather than adding. More importantly, the presentation of the results is not as clear as it should be. There are a lot of stimulus combinations, and they're not illustrated or described with the clarity that is needed. In summary, it's an interesting and novel paper, but presented in a way that pitches the results as something that is unclear that they're really related to, and doesn't do so in a clear fashion.

1) Calling this attention is a stretch. Attention is an internal state, and the only way to define it is through the subject's behavior (improved accuracy and/or speed). In this case no such behavioral changes can be observed, so calling this "attention" seems unjustified. Now a fair case could be made that attention is essentially prediction. In this work, prediction is quite evident, and this is essentially the justification the authors give for calling it attention. But if we go that far, then direction selectivity is also attention. I would say this is a very intriguing mechanism, and very nice work by the authors, but putting the attention label on it does not do it a service.

2) Now, the important question here is what the authors have shown once the attention veneer is removed. These are novel studies. However, direction selectivity is a common phenomenon observed across mammalian species in cortex, superior colliculus, and retina. So what's really new in this case? Most studies of direction selectivity focus on the receptive field, and don't consider the affect of trajectory across the whole visual field. Davies et al. (Scientific Reports, 2016) use Gabor patches presented to marmoset MT and find trajectory as a stimulus enters the RF is important in determining response, but their tests aren't as comprehensive as those presented here. Similarly, Priebe et al. tested speed tuning in MT (J Neurosci, 2003), although again this was focused on the RF. In primate MST, it's common to study large flow fields, which are more analogous to the stimuli here. None of this motion work, and many others, is really referenced here. And to my mind it's the context that this work belongs in, not attention.

3) Finally, a critical issue I have is that the description of the stimulus paradigm is confusing and not straightforward. Target probes are drifted along trajectories, and have orientations. The targets can be horizontally or vertically offset. Isn't that just a line? A dotted line? The little stimulus panels in Figure 1 don't really help. I think those are arrows to indicate drift direction, but they look like long lines. In any case, it should be easier to figure out what's going on here. The paper would greatly benefit from a nicer introduction figure that couples the recording traces in Figure 1 with better stimulus diagrams and a definition of their terms. For instance, the terms "primer" and "probe" are used extensively throughout the manuscript, but not really introduced in a simple and clear way. It's more through inference that I figured out what they mean, rather than through the authors directly illustrating and describing them. Adding to this complexity are the probe configurations – it can jump backwards, pause, and be occluded. This begs for an illustrative figure. Maybe even linked movie files. I assume this is all straightforward to the authors – to the reader, it's confusing.

[Editors' note: further revisions were requested prior to acceptance, as described below.]

Thank you for submitting your article "A predictive focus of gain modulation encodes target trajectories in insect vision" for consideration by *eLife*. Your article has been reviewed by Fred Rieke as the Reviewing Editor and Timothy Behrens as the Senior Editor.

This paper has improved in revision, but a number of issues about clarity of presentation remain before a final decision can be reached. These should all be straightforward to deal with by modifications in the text or figures.

1) Introduction: the first sentence is very specific; I think a broader first sentence (and even first paragraph) would be more appropriate for a diverse audience.

2) Introduction, first paragraph: the last sentence is unclear.

3) Subsection “Receptive fields are modulated by stimulus history”, first paragraph: it would be helpful to briefly describe the previous findings about selective attention to provide context for the present paper.

4) Figure 1 could still be improved. Could you clarify exactly what is plotted in the pseudo color images – is this the firing rate as a function of location, start point, end point, or…? In addition, the increase of firing rate over several hundred ms is not clear from the figure. Can you plot instantaneous firing rate over time or expand part of the trace in B? Finally, the suppression of spontaneous activity (subsection “Receptive fields are modulated by stimulus history”, last sentence) is not clear as it is not clear what the spontaneous activity is.

5) Figure 3: The regions of enhancement and suppression should be backed up with statistical tests.

6)“Facilitating stimuli certainly increase the firing rate against a steadily hyperpolarizing membrane potential”: this statement is not clear; it also would be nice to illustrate the hyper polarization more clearly in the figure.

7) “As in attention in planned eye-movements in primates (Zirnsak et al. 2014; Xiao 319 et al. 2007) it is difficult to disambiguate its potential role in sensory, planning or motor control”: this sentence is awkward; specifically it is important to clarify what "its" in "its potential role" refers to.

8) Another observation that could make it into the Discussion is the "omitted stimulus response" described by Michael Berry and colleagues. This is again a bit more than simple adaptation or facilitation, and shows a form of predictive coding that is entrained to the stimulus statistics.

---

## [Author Response]

*All three reviewers agreed that the core results in the paper were interesting and that the experiments appeared to be well conceived and executed. The reviewers also agreed that several issues needed work before a complete evaluation of the paper could be made. Three issues in particular emerged from discussions among the reviewers. These are highlighted below, and are elaborated in comments from the individual reviewers.*

*1) Attention vs. prediction*

*A key point emphasized in the reviews is that the phenomena described do not clearly satisfy the requirements for attentional effects. Specifically, attention is defined relative to behavior and not neural responses. The effect described in the paper would seem better described as a "predictive gain enhancement" or "anticipatory coding" or something similar. See comments from reviewers 2 and 3 for more details.*

Interestingly, in our 2013 Current Biology article detailing responses to two targets, reviewers stated that we should change our terminology from ‘competitive selection’ to ‘selective attention’. Here, we are again happy to place our research within the terminology suggested by the reviewers.

We observe *both excitation and suppression* (differentiating from the retina work and other direction selectivity studies), therefore we now use the term *gain modulation*, rather than enhancement. Rather than evoke controversy with the spotlight term, we refer to a ‘focus’ region.

Anticipation in the retina accounts for delays, but does not *predict* future locations, i.e. in the retina following straight line movement, sensitization is observed at the last seen location. Our enhancement spreads further forward, the longer the occlusion, thus predicting future target locations at corresponding future times (again differentiating this from models of direction selectivity).

We have therefore changed our terminology to ‘predictive focus of gain modulation’ and made these distinctions clearer in both the Introduction and Discussion.

*2) Novelty and relation to direction selectivity literature*

*There is a large literature on direction selectivity – including some in retina which considers complex stimulus trajectories (particularly work from Michael Berry – e.g. Schwartz et al., 2007; Chen et al., 2013; Chen et al., 2014). This work reveals a number of non-classical features of directional coding. This literature seems the proper context in which to evaluate the novelty of the present work, and the paper would benefit from comparing the work with what is known about direction selectivity in other systems.*

Our results are quite distinct from the anticipatory effects observed in the vertebrate retina or classic ‘elementary motion detector’ (EMD) models (e.g. Hassenstein-Reichardt Correlator). Our results include several observations that cannot be accounted for by classical direction-selective models including:

A) enhancement that spreads further forward, the longer the occlusion (prediction);

B) direction selectivity is itself changed by the direction of past trajectory;

C) facilitatory effects on longer durations (hundreds rather than tens of milliseconds) and much larger spatial scale (tens of degrees), compared with insect EMD retinotopic processing (~1-2°);

D) a suppressive surround over a very large spatial (~50°) extent and longer time scales (> 500 ms);

E) a contrast invariant ‘switch’ (recruited by either low or high contrast primers) rather than an activity-dependent mechanism, with either inducing the same shift in the contrast sensitivity function;

F) transfer of a local, predictive focus across brain hemispheres.

We have now described these points in finer detail, in either the Introduction or Discussion.

Additionally, a recent phenomenon observed in vertebrate retina is a population discharge of activity when a stimulus trajectory reverses. We observe the opposite, with a reversal evoking inhibition rather than excitation, thus our result encodes path continuation. We have added Figure 6 to further emphasize this strong inhibition due to a trajectory reversal.

*3) Presentation*

*Several issues made the paper harder to read than necessary. Most notably, the figure panels used to illustrate the stimuli were less helpful than they could be. It is essential to make this as clear as possible to the reader. See reviewer comments for specifics.*

We have made many text and figure modifications to enhance the presentation, including the development of stimulus videos. We have modified illustrative pictograms for clarity and expanded and improved description in text. We have included ‘primer’, ‘probe’ and ‘primer & probe’ examples in Figure 1, including zoomed-in illustrations of responses in both Figure 1 and 2. Further details of these changes are described below and tracked in the document.

*Reviewer #1:*

*In this paper, Wiederman and colleagues describe an interesting property of responses of visual neurons in dragonfly in which the sensitivity is enhanced to predict the trajectory of moving objects. I am not an expert in the main areas covered by the paper, but found the phenomena described quite interesting. There were a few places I felt the results could be presented more clearly:*

*The Results would likely to be easier to read with titled subsections that provide an outline of how the results are organized.*

We have added titled results subsections.

*Figure 1) Indicate which stimuli correspond to the example traces shown. 2) Why are some receptive fields smoothed and others not?*

We have included additional stimuli (Figure 1) linking the stimulus to example response traces. The pictograms are illustrative of the experimental design and we have noted this in text. We have clarified pictograms with a single stimulus and single data trace. We take the reader through the analysis steps of binning the data before interpolation which smooths edge artefacts of the binning process. This permits a more visually clear and scientifically sound mapping of the predictive gain modulation in Figure 3.

*Figure 2: Can you illustrate the differences in responses with and without the primer more clearly (e.g. blow up responses at the time of interest, perhaps give spike count as well)? In the present format it is not possible to see the differences in response, which means that Figure 2 supports Figure 3 less than it could.*

We have modified the figures to include enlarged versions and spike count.

*Figure 5: Can you make the stimulus in B clearer or include a bit more in the figure legend?*

We have changed the figure legend of Figure 5 to clearly describe the changing stimulus parameters for the example traces.

“Example data traces of responses to either low (grey) or high (black) contrast primers that are presented before a series of varying contrast probes (light, medium and dark blue)”.

*The Discussion is quite brief. If possible, the Discussion could be strengthened by making more connections to the broader attention literature. For example, the specificity of the enhanced sensitivity shown here, and the shift in direction tuning, are quite interesting. Is there precedent in other systems for those?*

We have changed our focus from attention to predictive gain modulation. We have extended the Discussion to include two new sections: (1) differences between our observations and direction selective models. (2) placing these results in context to our previous selective attention result (as suggested by reviewer 2).

*Please include significance tests, e.g. for the data in Figure 3.*

We have added a statistical test for Figure 3, testing the spreading forward enhancement in relation to the duration of the occlusion.

*Reviewer #2:*

*[…] 1) In the last sentence of the first paragraph, the authors suggest that retinal processing "does not predict an object's future location." However, in Kastner and Baccus, cited one sentence before, those authors state, "Sensitization acts to predict the future location of an object". The authors should clarify and expand their claims about what the retinal literature says and how it relates to the effects measured here.*

The retinal processing ‘predicts’ in the sense that for the current time (at 0 ms) it enhances where a target should be now given a straight trajectory (accounting for neuronal delay). However, our system predicts where the target could be at many future locations and times (over hundreds of milliseconds). We have modified our Introduction to describe this effect more clearly – that given an occlusion (remove the stimulus for a duration) the retina work would not predict where the target will be into the future.

*2) The largest question I have about this work is whether the observed effects constitute an "attentional spotlight", as the authors contend, or whether it is a nice example of non-linear stimulus-stimulus interactions over space and time, which would not typically be thought of as attention.*

*A) Attention can be a squirrelly concept, but I think a prime place to look is in the retina literature, which seems not to have adopted the term "attention" for similar phenomenology, in which anticipatory coding is observed (Kastner and Baccus, 2013; Schwartz… Berry, 2007). Are the retina folks just being too timid in their interpretation of their data? The term favored here, 'spotlight of attention', appears to most often refer to top-down attention. For typical experiments showing top-down attention, experimenters present a cue to guide where gain modulation is focused, but crucially, that cue should be rather separate from the probe stimulus. (For instance, with moving colored dots, a cue to indicate whether color or direction is the important variable for a given trial's reward.) Without a behavior in these experiments, it seems very difficult to argue for top-down attention. The authors rightly view the gain changes they measure as stimulus driven – they refer to a primer stimulus throughout.*

*B) Bottom-up attention is much harder to distinguish from stimulus-stimulus interactions, and seems less amenable to the term "attentional spotlight". That term has been used to apply to serial searches in human psychophysics where stimuli do not 'pop out' (in Anne Treisman's work in the 1980s). In those experiments, the attentional spotlight is in contrast to the popping out of stimuli, and is inferred from the serial observation of each object in a set.*

*C) Note also that the human reference (Watamaniuk and McKee) for this type cognitive ability (tracking continuous motion in the face of occlusion) does not refer to attention. In the primate and human studies of attention cited in the discussion (Reynolds et al., 2000 and Carrasco et al., 2000), the visual cues that were used to direct the covert attention of the subject were quite distinct from probe stimulus; that seems very different from the priming cues in this study, which were short dot motions closely related to the longer probing stimulus.*

*D) The authors' previous work (Current Biology, 2013) refers to stimulus selection by this neuron as "selective attention", but that seems less loaded than "attentional spotlight". A spotlight of attention could exist in invertebrates; I'm just not sure this study shows evidence for it.*

We have changed our terminology to a “predictive focus of gain modulation”. For the reviewers’ interest, we have worked out a method to determine which target is being selected at any instant and are currently studying whether our prediction and selection is exogenously or endogenously driven. However, we agree that our observation of a predictive ‘focus’ is itself of significant interest although it does not yet necessarily represent ‘attention’ as defined.

*3) In Figure 3 and results presentation (Results, fifth paragraph), the authors suggest that the spread forward in gain is further when the delay is longer. By eye, this does not look significant, but the authors should test for significance to make this claim. It would also be nice to note the extrapolated position from the priming motion after the 100 ms and 300 ms pauses.*

We have added significance tests and describe positions in the legend (3 and 9 degrees).

*4) These authors previously reported that this neuron's activity switched back and forth between the activity related to the trajectory of just one of two simultaneously presented targets (Wiederman & O'Carroll, 2013). The anticipatory gain control measured here gives a beautiful explanation for positive feedback that could cause the neuron to lock onto a single object while suppressing competing objects. I think this would be a relevant point to add the Discussion, and could also be used to tie these effects back to dragonfly ethology.*

We agree that it is beautiful. We have added the points raised by the reviewer to the Discussion (positive feedback, locking on).

*Reviewer #3:*

*[…] Overall, this work is quite interesting and shows that these particular neurons in the dragonfly brain are doing some remarkable computation. The authors frame this work in the context of attention. This I find to be a dubious choice, and I think detracts from the work rather than adding. More importantly, the presentation of the results is not as clear as it should be. There are a lot of stimulus combinations, and they're not illustrated or described with the clarity that is needed. In summary, it's an interesting and novel paper, but presented in a way that pitches the results as something that is unclear that they're really related to, and doesn't do so in a clear fashion.*

We have re-framed our results to reduce the emphasis on attention. We have enhanced our presentation of data and description of the stimuli, including adding video illustrations of the stimuli.

*1) Calling this attention is a stretch. Attention is an internal state, and the only way to define it is through the subject's behavior (improved accuracy and/or speed). In this case no such behavioral changes can be observed, so calling this "attention" seems unjustified. Now a fair case could be made that attention is essentially prediction. In this work, prediction is quite evident, and this is essentially the justification the authors give for calling it attention. But if we go that far, then direction selectivity is also attention. I would say this is a very intriguing mechanism, and very nice work by the authors, but putting the attention label on it does not do it a service.*

We thank the reviewer for their support of our work and have modified the manuscript away from emphasizing attention in this form.

*2) Now, the important question here is what the authors have shown once the attention veneer is removed. These are novel studies. However, direction selectivity is a common phenomenon observed across mammalian species in cortex, superior colliculus, and retina. So what's really new in this case? Most studies of direction selectivity focus on the receptive field, and don't consider the affect of trajectory across the whole visual field. Davies et al. (Scientific Reports, 2016) use Gabor patches presented to marmoset MT and find trajectory as a stimulus enters the RF is important in determining response, but their tests aren't as comprehensive as those presented here. Similarly, Priebe et al. tested speed tuning in MT (J Neurosci, 2003), although again this was focused on the RF. In primate MST, it's common to study large flow fields, which are more analogous to the stimuli here. None of this motion work, and many others, is really referenced here. And to my mind it's the context that this work belongs in, not attention.*

We have added a paragraph in the Discussion describing what differentiates our results from models of direction selectivity. We have added references to direction selective models in this comparison. Davies et al. examine how a trajectory improves the encoding of speed in MT neurons, which we believe is quite different to the predictive gain modulation to moving features that we present here.

*3) Finally, a critical issue I have is that the description of the stimulus paradigm is confusing and not straightforward. Target probes are drifted along trajectories, and have orientations. The targets can be horizontally or vertically offset. Isn't that just a line? A dotted line? The little stimulus panels in Figure 1 don't really help. I think those are arrows to indicate drift direction, but they look like long lines. In any case, it should be easier to figure out what's going on here. The paper would greatly benefit from a nicer introduction figure that couples the recording traces in Figure 1 with better stimulus diagrams and a definition of their terms. For instance, the terms "primer" and "probe" are used extensively throughout the manuscript, but not really introduced in a simple and clear way. It's more through inference that I figured out what they mean, rather than through the authors directly illustrating and describing them. Adding to this complexity are the probe configurations – it can jump backwards, pause, and be occluded. This begs for an illustrative figure. Maybe even linked movie files. I assume this is all straightforward to the authors – to the reader, it's confusing.*

We have developed new insets into Figure 1 to further develop the paradigm behind primer and probe (simpler and more clearly). We have changed pictograms and added in text better descriptions of the target as a small, black square. We have added video files as additional descriptions of the stimuli.

[Editors' note: further revisions were requested prior to acceptance, as described below.]

*This paper has improved in revision, but a number of issues about clarity of presentation remain before a final decision can be reached. These should all be straightforward to deal with by modifications in the text or figures.*

*1) Introduction: the first sentence is very specific; I think a broader first sentence (and even first paragraph) would be more appropriate for a diverse audience.*

We have modified the opening sentences to more generally introduce the topic.

2) Introduction, first paragraph: the last sentence is unclear.

We have re-worded this sentence, expanding it to make it clearer.

“This differs from studies of human observers, where a temporarily occluded target results in improved sensitivity at the extrapolated forward location (Watamaniuk & McKee 1995). This predictive encoding of future target locations indicates the presence of additional processing mechanisms beyond the retina."

3) Subsection “Receptive fields are modulated by stimulus history”, first paragraph: it would be helpful to briefly describe the previous findings about selective attention to provide context for the present paper.

We have added the following description (including an additional reference).

“One identified STMD, CSTMD1, responds selectively to a small, moving target, even when embedded within natural scenes (Wiederman & O’Carroll, 2011). CSTMD1 also exhibits a sophisticated form of selective attention. […] Instead, CSTMD1 responds in a winner-takes-all manner, selecting a single target as if the distracter does not even exist (Wiederman & O’Carroll, 2013).”

*4) Figure 1 could still be improved. Could you clarify exactly what is plotted in the pseudo color images – is this the firing rate as a function of location, start point, end point, or…? In addition, the increase of firing rate over several hundred ms is not clear from the figure. Can you plot instantaneous firing rate over time or expand part of the trace in B? Finally, the suppression of spontaneous activity (subsection “Receptive fields are modulated by stimulus history”, last sentence) is not clear as it is not clear what the spontaneous activity is.*

We have added the analysis window here, that was previously in the Materials and methods section. We have added time courses to the figure (mean over many replicates), to show the response onsets to the two conditions (probe alone, primer and probe). This figure addition of response time courses is described in text and legend. We have calculated spontaneous activity levels and included this in text.

*5) Figure 3: The regions of enhancement and suppression should be backed up with statistical tests.*

This is a great suggestion and we apologise we missed covering it in the first review stage. How do we illustrate that our focus regions in Figure 1 are not simply noise (besides the statistical verification in Figure 3 (now E) and Figure 4)? To address this, we derived new map versions (Figure 1) that illustrate the *effect size*. That is, the Cohen’s d maps indicate how large the change is (primer & probe – probe alone), in comparison to the inter-sample (dragonflies) variability for each spatial location (standard deviation). The effect sizes we report are very large – up to 1.8 in both excitatory and inhibitory directions (>0.5 traditionally considered large). We also took the opportunity to apply statistical tests at spatial location we note in text, e.g. “over 50% increases”. We write the P value at these locations on our Cohen’s d maps to reveal the statistical significance of our assertions.

*6)“Facilitating stimuli certainly increase the firing rate against a steadily hyperpolarizing membrane potential”: this statement is not clear; it also would be nice to illustrate the hyper polarization more clearly in the figure.*

We are at the disadvantage that our spiking responses in CSTMD1 are so large (~80mV), therefore a 4mV hyperpolarization is difficult to illustrate. However, we added a new figure showing this hyperpolarization in a data trace and we believe it is clear.

*7) “As in attention in planned eye-movements in primates (Zirnsak et al. 2014; Xiao 319 et al. 2007) it is difficult to disambiguate its potential role in sensory, planning or motor control”: this sentence is awkward; specifically it is important to clarify what "its" in "its potential role" refers to.*

This sentence should have been erased when we changed the focus away from attention, and towards predictive gain modulation. We simplified to the following.

“CSTMD1 is a higher-order neuron with inputs in the anterior optic tubercle, a midbrain output destination of optic lobe interneurons. We also tested for the facilitatory component of the predictive gain modulation in likely pre-cursor neurons: small-field (SF) STMDs located at an earlier stage of visual processing (Barnett et al., 2007).”

*8) Another observation that could make it into the Discussion is the "omitted stimulus response" described by Michael Berry and colleagues. This is again a bit more than simple adaptation or facilitation, and shows a form of predictive coding that is entrained to the stimulus statistics.*

We added this point in the Discussion, just where your comment alludes to – “more complex than simple neuronal adaptation, sensitization, habituation or fatigue”.

*“*Here modulation of the dynamic receptive field represents anticipatory coding, a more complex influence of history than simple neuronal adaptation, sensitization, habituation or fatigue. […] Instead of encoding novelty or the unexpected, STMD neurons predict consistency of a selected target’s trajectory, all whilst suppressing distracters.”